# Structured Convolutions for Efficient Neural Network Design

**Yash Bhalgat**      **Yizhe Zhang**      **Jamie Menjay Lin**      **Fatih Porikli**

Qualcomm AI Research*
{ybhalgat, yizhez, jmlin, fporikli}@qti.qualcomm.com

## Abstract

In this work, we tackle model efficiency by exploiting redundancy in the *implicit structure* of the building blocks of convolutional neural networks. We start our analysis by introducing a general definition of Composite Kernel structures that enable the execution of convolution operations in the form of efficient, scaled, sum-pooling components. As its special case, we propose *Structured Convolutions* and show that these allow decomposition of the convolution operation into a sum-pooling operation followed by a convolution with significantly lower complexity and fewer weights. We show how this decomposition can be applied to 2D and 3D kernels as well as the fully-connected layers. Furthermore, we present a Structural Regularization loss that promotes neural network layers to leverage on this desired structure in a way that, after training, they can be decomposed with negligible performance loss. By applying our method to a wide range of CNN architectures, we demonstrate 'structured' versions of the ResNets that are up to $2\times$ smaller and a new Structured-MobileNetV2 that is more efficient while staying within an accuracy loss of 1% on ImageNet and CIFAR-10 datasets. We also show similar structured versions of EfficientNet on ImageNet and HRNet architecture for semantic segmentation on the Cityscapes dataset. Our method performs equally well or superior in terms of the complexity reduction in comparison to the existing tensor decomposition and channel pruning methods.

## 1 Introduction

Deep neural networks deliver outstanding performance across a variety of use-cases but quite often fail to meet the computational budget requirements of mainstream devices. Hence, model efficiency plays a key role in bridging deep learning research into practice. Various model compression techniques rely on a key assumption that the deep networks are over-parameterized, meaning that a significant proportion of the parameters are redundant. This redundancy can appear either explicitly or implicitly. In the former case, several structured [12, 23], as well as unstructured [8, 9, 27, 46], pruning methods have been proposed to systematically remove redundant components in the network and improve run-time efficiency. On the other hand, tensor-decomposition methods based on singular values of the weight tensors, such as spatial SVD or weight SVD, remove somewhat implicit elements of the weight tensor to construct low-rank decompositions for efficient inference [5, 14, 19].

Redundancy in deep networks can also be seen as network weights possessing an unnecessarily high degrees of freedom (DOF). Alongside various regularization methods [17, 33] that impose constraints to avoid overfitting, another approach for reducing the DOF is by decreasing the number of *learnable* parameters. To this end, [14, 29, 38] propose using certain basis representations for weight tensors.

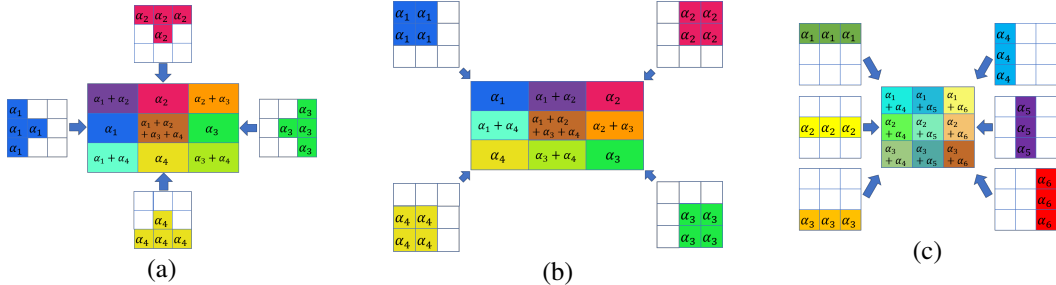

Figure 1: A $3 \times 3$ composite kernel constructed as a superimposition of different underlying structures. Kernels in (a) and (b) possess 4 degrees of freedom whereas the kernel in (c) has 6 degrees of freedom. *Color combinations are chosen to reflect the summations, this figure is best viewed in color.*

In these methods, the basis vectors are fixed and only their coefficients are learnable. Thus, by using a smaller number of coefficients than the size of weight tensors, the DOF can be effectively restricted. But, note that, this is useful only during training since the original higher number of parameters are used during inference. [29] shows that systematically choosing the basis (e.g. the Fourier-Bessel basis) can lead to model size shrinkage and flops reduction even during inference.

In this work, we explore restricting the degrees of freedom of convolutional kernels by imposing a structure on them. This structure can be thought of as constructing the convolutional kernel by super-imposing several *constant-height* kernels. A few examples are shown in Fig. 1, where a kernel is constructed via superimposition of $M$ linearly independent masks with associated constant scalars $\alpha_m$, hence leading to $M$ degrees of freedom for the kernel. The very nature of the basis elements as binary masks enables efficient execution of the convolution operation as explained in Sec. 3.1.

In Sec. 4, we introduce *Structured Convolutions* as a special case of this superimposition and show that it leads to a decomposition of the convolution operation into a sum-pooling operation and a significantly smaller convolution operation. We show how this decomposition can be applied to convolutional layers as well as fully connected layers. We further propose a regularization method named *Structural Regularization* that promotes the normal convolution weights to have the desired structure that facilitates our proposed decomposition. Overall, our key contributions in this work are:

1. We introduce Composite Kernel structure, which accepts an arbitrary basis in the kernel formation, leading to an efficient convolution operation. Sec. 3 provides the definition.

2. We propose Structured Convolutions, a realization of the composite kernel structure. We show that a structured convolution can be decomposed into a sum-pooling operation followed by a much smaller convolution operation. A detailed analysis is provided in Sec. 4.1.

3. Finally, we design Structural Regularization, an effective training method to enable the structural decomposition with minimal loss of accuracy. Our process is described in Sec. 5.1.

## 2 Related Work

The existing literature on exploiting redundancy in deep networks can be broadly studied as follows.

**Tensor Decomposition Methods.** The work in [47] proposed a Generalized SVD approach to decompose a $B \times C \times k \times k$ convolution (where $B$ and $C$ are output and input channels, and $k$ is the spatial size) into a $B' \times C \times k \times k$ convolution followed by a $B \times B' \times 1 \times 1$ convolution. Likewise, [14] introduced Spatial SVD to decompose a $k \times k$ kernel into $k \times p$ and $p \times k$ kernels. [36] further developed a non-iterative method for such low-rank decomposition. CP-decomposition [15, 20] and tensor-train decomposition [28, 35, 43] have been proposed to decompose high dimensional tensors. In our method, we too aim to decompose regular convolution into computationally lightweight units.

**Structured Pruning.** [11, 12, 22] presented channel pruning methods where redundant channels in every layer are removed. The selection process of the redundant channels is unique to every method, for instance, [12] addressed the channel selection problem using lasso regression. Similarly, [41] used group lasso regularization to penalize and prune unimportant groups on different levels of granularity. We refer readers [19] for a survey of structured pruning and tensor decomposition methods. To

our advantage, the proposed method in this paper does not explicitly prune, instead, our structural regularization loss imposes a form on the convolution kernels.

**Semi-structured and Unstructured Pruning.** Other works [21, 25, 6] employed block-wise sparsity (also called *semi-structured* pruning) which operates on a finer level than channels. Unstructured pruning methods [1, 8, 18, 46] prune on the parameter-level yielding higher compression rates. However, their unstructured nature makes it difficult to deploy them on most hardware platforms.

**Using Prefixed Basis.** Several works [29, 38] applied basis representations in deep networks. Seminal works [26, 32] used wavelet bases as feature extractors. Choice of the basis is important, for example, [29] used Fourier-Bessel basis that led to a reduction in computation complexity. In general, tensor decomposition can be seen as basis representation learning. We propose using structured binary masks as our basis, which leads to an immediate reduction in the number of multiplications.

Orthogonal to structured compression, [24, 42, 49] utilized shift-based operations to reduce the overall computational load. Given the high computational cost of multiplications compared to additions [13], [3] proposed networks where the majority of the multiplications are replaced by additions.

## 3    Composite Kernels

We first give a definition that encompasses a wide range of structures for convolution kernels.

**Definition 1.** *For $\mathcal{R}^{C \times N \times N}$, a Composite Basis $\mathcal{B} = \{\beta_1, \beta_2, ..., \beta_M\}$ is a linearly independent set of binary tensors of dimension $C \times N \times N$ as its basis elements. That is, $\beta_m \in \mathcal{R}^{C \times N \times N}$, each element $\beta_{mijk}$ of $\beta_m \in \{0, 1\}$, and $\sum_{m=1}^{M} \alpha_m \beta_m = 0$ iff $\alpha_m = 0 \ \forall m$.*

The linear independence condition implies that $M \leq CN^2$. Hence, the basis spans a subspace of $\mathcal{R}^{C \times N \times N}$. The speciality of the Composite Basis is that the basis elements are binary, which leads to an immediate reduction in the number of multiplications involved in the convolution operation.

**Definition 2.** *A kernel $W \in \mathcal{R}^{C \times N \times N}$ is a Composite Kernel if it is in the subspace of the Composite Basis. That is, it can be constructed as a linear combination of the elements of $\mathcal{B}$: $\exists \ \boldsymbol{\alpha} = [\alpha_1, .., \alpha_M]$ such that $W := \sum_{m=1}^{M} \alpha_m \beta_m$.*

Note that, the binary structure of the underlying Composite Basis elements defines the structure of the Composite Kernel. Fig. 1 shows a $3 \times 3$ Composite Kernel (with $C = 1, N = 3$) constructed using different examples of a Composite Basis. In general, the underlying basis elements could have a more random structure than what is demonstrated in those examples shown in Fig. 1.

Conventional kernels (with no restrictions on DOF) are just special cases of Composite Kernels, where $M = CN^2$ and each basis element has only one nonzero element in its $C \times N \times N$ grid.

### 3.1    Convolution with Composite Kernels

Consider a convolution with a Composite Kernel of size $C \times N \times N$, where $N$ is the spatial size and $C$ is the number of input channels. To compute an output, this kernel is convolved with a $C \times N \times N$ volume of the input feature map. Let's call this volume $X$. Therefore, the output at this point will be:

$$X * W = X * \sum_{m=1}^{M} \alpha_m \beta_m = \sum_{m=1}^{M} \alpha_m (X * \beta_m) = \sum_{m=1}^{M} \alpha_m \, sum(X \bullet \beta_m) = \sum_{m=1}^{M} \alpha_m E_m \quad (1)$$

where '$*$' denotes convolution, '$\bullet$' denotes element-wise multiplication. Since $\beta_m$ is a binary tensor, $sum(X \bullet \beta_m)$ is same as adding the elements of $X$ wherever $\beta_m = 1$, thus no multiplications are needed. Ordinarily, the convolution $X * W$ would involve $CN^2$ multiplications and $CN^2 - 1$ additions. In our method, we can trade multiplications with additions. From (1), we can see that we only need $M$ multiplications and the total number of additions becomes:

$$\text{Num additions} = \sum_{m=1}^{M} \underbrace{(sum(\beta_m) - 1)}_{\text{from } sum(X \bullet \beta_m)} + \underbrace{(M - 1)}_{\text{from } \sum \alpha_m E_m} = \sum_{m=1}^{M} sum(\beta_m) - 1 \quad (2)$$

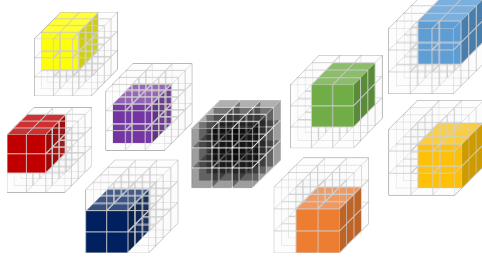

**Figure 2:** $4 \times 3 \times 3$ structured kernel constructed with 8 basis elements each having a $3 \times 2 \times 2$ patch of 1's. *Figure best viewed in color.*

**Figure 3:** $3 \times 3$ structured convolution is equivalent to $2 \times 2$ sum-pooling + $2 \times 2$ convolution.

Depending on the structure, number of the additions *per output* can be larger than $CN^2 - 1$. For example, in Fig. 1(b) where $C = 1, N = 3, M = 4$, we have $\sum_m sum(\beta_m) - 1 = 15 > CN^2 - 1 = 8$). In Sec. 4.2, we show that the number of additions can be amortized to as low as $M - 1$.

## 4 Structured Convolutions

**Definition 3.** *A kernel in $\mathcal{R}^{C \times N \times N}$ is a Structured Kernel if it is a Composite Kernel with $M = cn^2$ for some $1 \le n \le N, 1 \le c \le C$, and if each basis tensor $\beta_m$ is made of a $(C - c + 1) \times (N - n + 1) \times (N - n + 1)$ cuboid of 1's, while rest of its coefficients being $0$.*

A Structured Kernel is characterized by its dimensions $C, N$ and its underlying parameters $c, n$. Convolutions performed using Structured Kernels are called Structured Convolutions.

Fig. 1(b) depicts a 2D case of a $3 \times 3$ structured kernel where $C = 1, N = 3, c = 1, n = 2$. As shown, there are $M = cn^2 = 4$ basis elements and each element has a $2 \times 2$ sized patch of 1's.

Fig. 2 shows a 3D case where $C = 4, N = 3, c = 2, n = 2$. Here, there are $M = cn^2 = 8$ basis elements and each element has a $3 \times 2 \times 2$ cuboid of 1's. Note how these cuboids of 1's (shown in colors) cover the entire $4 \times 3 \times 3$ grid.

### 4.1 Decomposition of Structured Convolutions

A major advantage of defining Structured Kernels this way is that all the basis elements are just shifted versions of each other (see Fig. 2 and Fig. 1(b)). This means, in Eq. (1), if we consider the convolution $X * W$ for the *entire* feature map $X$, the summed outputs $X * \beta_m$ for all $\beta_m$'s are actually the same (except on the edges of $X$). As a result, the outputs $\{X * \beta_1, ..., X * \beta_{cn^2}\}$ can be computed using a *single sum-pooling operation* on $X$ with a kernel size of $(C - c + 1) \times (N - n + 1) \times (N - n + 1)$. Fig. 3 shows a simple example of how a convolution with a $3 \times 3$ structured kernel can be broken into a $2 \times 2$ sum-pooling followed by a $2 \times 2$ convolution with a kernel made of $\alpha_i$'s.

Furthermore, consider a convolutional layer of size $C_{out} \times C \times N \times N$ that has $C_{out}$ kernels of size $C \times N \times N$. In our design, the same underlying basis $\mathcal{B} = \{\beta_1, ..., \beta_{cn^2}\}$ is used for the construction of all $C_{out}$ kernels in the layer. Suppose any two structured kernels in this layer with coefficients $\boldsymbol{\alpha}^{(1)}$ and $\boldsymbol{\alpha}^{(2)}$, i.e. $W_1 = \sum_m \alpha_m^{(1)} \beta_m$ and $W_2 = \sum_m \alpha_m^{(2)} \beta_m$. The convolution output with these two kernels is respectively, $X * W_1 = \sum_m \alpha_m^{(1)} (X * \beta_m)$ and $X * W_2 = \sum_m \alpha_m^{(2)} (X * \beta_m)$. We can see that the computation $X * \beta_m$ is common to all the kernels of this layer. Hence, the sum-pooling operation only needs to be computed once and then reused across all the $C_{out}$ kernels.

A Structured Convolution can thus be decomposed into a sum-pooling operation and a smaller convolution operation with a kernel composed of $\alpha_i$'s. Fig. 4 shows the decomposition of a general structured convolution layer of size $C_{out} \times C \times N \times N$.

Notably, standard convolution ($C \times N \times N$), depthwise convolution ($1 \times N \times N$), and pointwise convolution ($C \times 1 \times 1$) kernels can all be constructed as 3D structured kernels, which means that this decomposition can be widely applied to existing architectures. See supplementary material for more details on applying the decomposition to convolutions with arbitrary *stride, padding, dilation.*

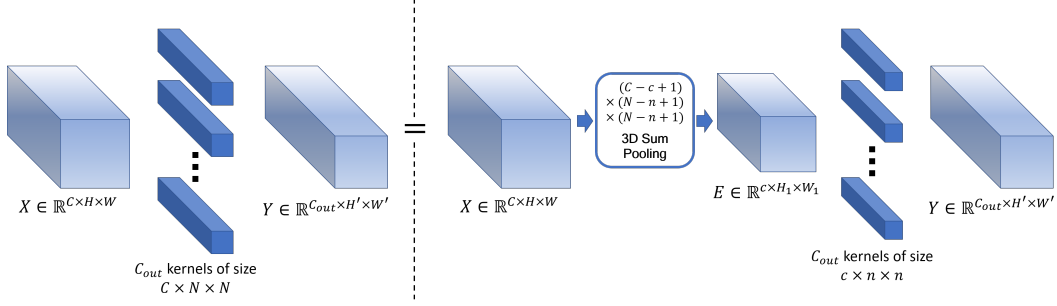

Figure 4: Decomposition of Structured Convolution. On the left, the conventional operation of a structured convolutional layer of size $C_{out} \times C \times N \times N$ is shown. On the right, we show that it is equivalent to performing a 3D sum-pooling followed by a convolutional layer of size $C_{out} \times c \times n \times n$.

## 4.2 Reduction in Number of Parameters and Multiplications/Additions

The sum-pooling component after decomposition requires no parameters. Thus, the total number of parameters in a convolution layer get reduced from $C_{out}CN^2$ (before decomposition) to $C_{out}cn^2$ (after decomposition). The sum-pooling component is also free of multiplications. Hence, only the smaller $c \times n \times n$ convolution contributes to multiplications after decomposition.

Before decomposition, computing every output element in feature map $Y \in \mathcal{R}^{C_{out} \times H' \times W'}$ involves $CN^2$ multiplications and $CN^2 - 1$ additions. Hence, total multiplications involved are $CN^2 C_{out} H' W'$ and total additions involved are $(CN^2 - 1)C_{out} H' W'$.

After decomposition, computing every output element in feature map $Y$ involves $cn^2$ multiplications and $cn^2 - 1$ additions. Hence, total multiplications and additions involved in computing $Y$ are $cn^2 C_{out} H' W'$ and $(cn^2 - 1)C_{out} H' W'$ respectively. Now, computing every element of the intermediate sum-pooled output $E \in \mathcal{R}^{c \times H_1 \times W_1}$ involves $((C - c + 1)(N - n + 1)^2 - 1)$ additions. Hence, the overall total additions involved can be written as:

$$C_{out} \left( \frac{((C - c + 1)(N - n + 1)^2 - 1)cH_1W_1}{C_{out}} + (cn^2 - 1)H'W' \right)$$

We can see that the number of parameters and number of multiplications have both reduced by a factor of $\mathbf{cn^2/CN^2}$. And in the expression above, if $C_{out}$ is large enough, the first term inside the parentheses gets amortized and the number of additions $\approx (cn^2 - 1)C_{out}H'W'$. As a result, the number of additions also reduce by approximately the same proportion $\approx cn^2/CN^2$. We will refer to $CN^2/cn^2$ as the compression ratio from now on.

Due to amortization, the additions *per output* are $\approx cn^2 - 1$, which is basically $M - 1$ since $M = cn^2$.

## 4.3 Extension to Fully Connected layers

For image classification networks, the last fully connected layer (sometimes called linear layer) dominates w.r.t. the number of parameters, especially if the number of classes is high. The structural decomposition can be easily extended to the linear layers by noting that a matrix multiplication is the same as performing a number of $1 \times 1$ convolutions on the input. Consider a kernel $W \in \mathcal{R}^{P \times Q}$ and input vector $X \in \mathcal{R}^{Q \times 1}$. The linear operation $WX$ is mathematically equivalent to the $1 \times 1$ convolution $unsqueezed(X) * unsqueezed(W)$, where $unsqueezed(X)$ is the same as $X$ but with dimensions $Q \times 1 \times 1$ and $unsqueezed(W)$ is the same as $W$ but with dimensions $P \times Q \times 1 \times 1$. In other words, each row of $W$ can be considered a $1 \times 1$ convolution kernel of size $Q \times 1 \times 1$.

Now, if each of these kernels (of size $Q \times 1 \times 1$) is *structured* with underlying parameter $R$ (where $R \leq Q$), then the matrix multiplication operation can be structurally decomposed as shown in Fig. 5.

Same as before, we get a reduction in both the number of parameters and the number of multiplications by a factor of $\mathbf{R/Q}$, as well as the number of additions by a factor of $\frac{R(Q-R)+(PR-1)P}{(PQ-1)P}$.

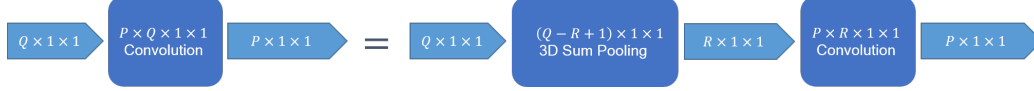

Figure 5: Structural decomposition of a matrix multiplication.

## 5    Imposing Structure on Convolution Kernels

To apply the structural decomposition, we need the weight tensors to be structured. In this section, we propose a method to impose the desired structure on the convolution kernels via training.

From the definition, $W = \sum_m \alpha_m \beta_m$, we can simply define matrix $A \in \mathcal{R}^{CN^2 \times cn^2}$ such that its $i^{th}$ column is the vectorized form of $\beta_i$. Hence, vectorized$(W) = A \cdot \boldsymbol{\alpha}$, where $\boldsymbol{\alpha} = [\alpha_1, ..., \alpha_{cn^2}]$.

Another way to see this is from structural decomposition. We may note that the $(C - c + 1) \times (N - n + 1) \times (N - n + 1)$ sum-pooling can also be seen as a convolution with a kernel of all 1's; we refer to this kernel as $\mathbf{1}_{(C-c+1) \times (N-n+1) \times (N-n+1)}$. Hence, the structural decomposition is:

$$X * W = X * \mathbf{1}_{(C-c+1) \times (N-n+1) \times (N-n+1)} * \boldsymbol{\alpha}_{c \times n \times n}$$

That implies, $W = \mathbf{1}_{(C-c+1) \times (N-n+1) \times (N-n+1)} * \boldsymbol{\alpha}_{c \times n \times n}$. Since the stride of the sum-pooling involved is 1, this can be written in terms of a matrix multiplication with a Topelitz matrix [34]:

$$\text{vectorized}(W) = \text{Topelitz}(\mathbf{1}_{(C-c+1) \times (N-n+1) \times (N-n+1)}) \cdot \text{vectorized}(\boldsymbol{\alpha}_{c \times n \times n})$$

Hence, the structure matrix $A$ referred above is basically $\text{Topelitz}(\mathbf{1}_{(C-c+1) \times (N-n+1) \times (N-n+1)})$.

### 5.1    Training with Structural Regularization

Now, for a structured kernel $W$ characterized by $\{C, N, c, n\}$, there exists a $cn^2$ length $\boldsymbol{\alpha}$ such that $W = A\boldsymbol{\alpha}$. Hence, a structured kernel $W$ satisfies the property: $W = AA^+W$, where $A^+$ is the Moore-Penrose inverse [2] of $A$. Based on this, we propose training a deep network with a Structural Regularization loss that can gradually push the deep network's kernels to be structured *via training*:

$$\mathcal{L}_{total} = \mathcal{L}_{task} + \lambda \sum_{l=1}^{L} \frac{\left\| (I - A_l A_l^+) W_l \right\|_F}{\|W_l\|_F} \tag{3}$$

where $\|\cdot\|_F$ denotes Frobenius norm and $l$ is the layer index. To ensure that regularization is applied uniformly to all layers, we use $\|W\|_F$ normalization in the denominator. It also stabilizes the performance of the decomposition w.r.t $\lambda$. The overall proposed training recipe is as follows:

**Proposed Training Scheme:**

- *Step 1*: Train the original architecture with the Structural Regularization loss.
  - After Step 1, all weight tensors in the deep network will be *almost* structured.
- *Step 2*: Apply the decomposition on every layer and compute $\alpha_l = A_l^+ W_l$.
  - This results in a smaller and more efficient decomposed architecture with $\alpha_l$'s as the weights. Note that, every convolution / linear layer from the original architecture is now replaced with a sum-pooling layer and a smaller convolution / linear layer.

The proposed scheme trains the architecture with the original $C \times N \times N$ kernels in place but with a structural regularization loss. The structural regularization loss imposes a restrictive $cn^2$ degrees of freedom while training but in a *soft* or gradual manner (depending on $\lambda$):

1. If $\lambda = 0$, it is the same as normal training with no structure imposed.
2. If $\lambda$ is very high, the regularization loss will be heavily minimized in early training iterations. Thus, the weights will be optimized in a restricted $cn^2$ dimensional subspace of $\mathcal{R}^{C \times N \times N}$.
3. Choosing a moderate $\lambda$ gives the best tradeoff between structure and model performance.

We talk about training implementation details for reproduction, such as hyperparameters and training schedules, in Supplementary material, where we also show our method is robust to the choice of $\lambda$.

# 6  Experiments

We apply structured convolutions to a wide range of architectures and analyze the performance and complexity of the decomposed architectures. We evaluate our method on ImageNet [30] and CIFAR-10 [16] benchmarks for image classification and Cityscapes [4] for semantic segmentation.

Table 1: Results: ResNets on CIFAR-10

| Architecture | Adds ($\times 10^6$) | Mults ($\times 10^6$) | Params ($\times 10^6$) | Acc. (in %) |
|---|---|---|---|---|
| ResNet56 | 125.49 | 126.02 | 0.85 | 93.03 |
| **Struct-56-A (ours)** | 63.04 | 62.10 | 0.40 | 92.65 |
| **Struct-56-B (ours)** | 48.49 | 33.87 | 0.21 | 91.78 |
| Ghost-Res56 [7] | 63.00 | 63.00 | 0.43 | 92.70 |
| ShiftRes56-6 [42] | 51.00 | 51.00 | 0.58 | 92.69 |
| AMC-Res56 [11] | 63.00 | 63.00 | – | 91.90 |
| ResNet32 | 68.86 | 69.16 | 0.46 | 92.49 |
| **Struct-32-A (ours)** | 35.59 | 35.09 | 0.22 | 92.07 |
| **Struct-32-B (ours)** | 26.29 | 18.18 | 0.11 | 90.24 |
| ResNet20 | 40.55 | 40.74 | 0.27 | 91.25 |
| **Struct-20-A (ours)** | 20.77 | 20.42 | 0.13 | 91.04 |
| **Struct-20-B (ours)** | 15.31 | 10.59 | 0.067 | 88.47 |
| ShiftRes20-6 [42] | 23.00 | 23.00 | 0.19 | 90.59 |

Table 2: Results: MobileNetV2 on ImageNet

| Architecture | Adds ($\times 10^6$) | Mults ($\times 10^6$) | Params ($\times 10^6$) | Acc. (in %) |
|---|---|---|---|---|
| MobileNetV2 | 0.30 | 0.31 | 3.50 | 72.19 |
| **Struct-V2-A (ours)** | 0.26 | 0.23 | 2.62 | 71.29 |
| **Struct-V2-B (ours)** | 0.29 | 0.17 | 1.77 | 64.93 |
| AMC-MV2 [11] | 0.21 | 0.21 | – | 70.80 |
| ChPrune-MV2-1.3x | 0.22 | 0.23 | 2.58 | 68.99 |
| Slim-MV2 [45] | 0.21 | 0.21 | 2.60 | 68.90 |
| WeightSVD 1.3x | 0.22 | 0.23 | 2.45 | 67.48 |
| ChPrune-MV2-2x | 0.15 | 0.15 | 1.99 | 63.82 |

Table 3: Results: ResNets on ImageNet

| Architecture | Adds ($\times 10^9$) | Mults ($\times 10^9$) | Params ($\times 10^6$) | Acc. (in %) |
|---|---|---|---|---|
| ResNet50 | 4.09 | 4.10 | 25.56 | 76.15 |
| **Struct-50-A (ours)** | 2.69 | 2.19 | 13.49 | 75.65 |
| **Struct-50-B (ours)** | 1.92 | 1.38 | 8.57 | 73.41 |
| ChPrune-R50-2x [12] | 2.04 | 2.05 | 17.89 | 75.44 |
| WeightSVD-R50 [48] | 2.04 | 2.05 | 13.40 | 75.12 |
| Ghost-R50 (s=2) [7] | 2.20 | 2.20 | 13.00 | 75.00 |
| Versatile-v2-R50 [40] | 3.00 | 3.00 | 11.00 | 74.50 |
| ShiftResNet50 [42] | – | – | 11.00 | 73.70 |
| Slim-R50 0.5x [45] | 1.10 | 1.10 | 6.90 | 72.10 |
| ResNet34 | 3.66 | 3.67 | 21.80 | 73.30 |
| **Struct-34-A (ours)** | 1.71 | 1.71 | 9.82 | 72.81 |
| **Struct-34-B (ours)** | 1.47 | 1.11 | 5.60 | 69.44 |
| ResNet18 | 1.81 | 1.82 | 11.69 | 69.76 |
| **Struct-18-A (ours)** | 0.88 | 0.89 | 5.59 | 69.13 |
| **Struct-18-B (ours)** | 0.79 | 0.63 | 3.19 | 66.19 |
| WeightSVD-R18 [48] | 0.89 | 0.90 | 5.83 | 67.71 |
| ChPrune-R18-2x [12] | 0.90 | 0.90 | 7.45 | 67.69 |
| ChPrune-R18-4x | 0.44 | 0.45 | 3.69 | 61.56 |

Entries are shortened, e.g. 'Channel Pruning' as 'ChPrune'. Results for [12, 48] are obtained from [19].

Table 4: Results: EfficientNet on ImageNet

| Architecture | Adds ($\times 10^6$) | Mults ($\times 10^6$) | Params ($\times 10^6$) | Acc. (in %) |
|---|---|---|---|---|
| EfficientNet-B1 [37] | 0.70 | 0.70 | 7.80 | 78.50[1] |
| **Struct-EffNet (ours)** | 0.62 | 0.45 | 4.93 | 76.40[1] |
| EfficientNet-B0 [37] | 0.39 | 0.39 | 5.30 | 76.10[1] |

## 6.1  Image Classification

We present results for ResNets [10] in Tables 1 and 3. To demonstrate the efficacy of our method on modern networks, we also show results on MobileNetV2 [31] and EfficientNet[2] [37] in Table 2 and 4.

To provide a comprehensive analysis, for each baseline architecture, we present *structured* counterparts, with version "A" designed to deliver similar accuracies and version "B" for extreme compression ratios. Using different $\{c, n\}$ configurations per-layer, we obtain structured versions with varying levels of reduction in model size and multiplications/additions (please see Supplementary material for details). For the "A" versions of ResNet, we set the compression ratio ($CN^2/cn^2$) to be $2\times$ for all layers. For the "B" versions of ResNets, we use nonuniform compression ratios per layer. Specifically, we compress stages 3 and 4 drastically ($4\times$) and stages 1 and 2 by $2\times$. Since MobileNet is already a compact model, we design its "A" version to be $1.33\times$ smaller and "B" version to be $2\times$ smaller.

We note that, on low-level hardware, additions are much power-efficient and faster than multiplications [3, 13]. Since the actual inference time depends on how software optimizations and scheduling are

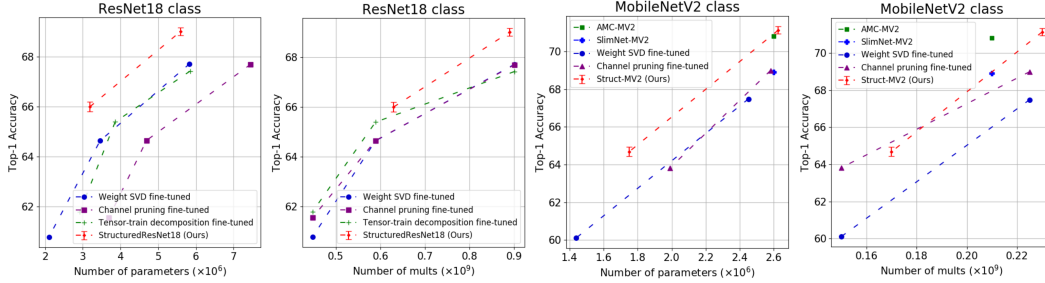

Figure 6: Comparison with structured compression methods shows 'ImageNet top-1 vs model size' trade-off as well as 'ImageNet top-1 vs multiplications' trade-off. Note that, we plot the **mean** $\pm$ **stddev** (error bars) values for our method, whereas the compared methods have provided their '**best**' numbers. Also, parameter count for AMC-MV2 is interpolated assuming uniform compression; the actual model would be larger than shown given that AMC [11] optimizes for MAC count.

implemented, for most objective conclusions, we provide the number of additions / multiplications and model sizes. Considering observations for sum-pooling on dedicated hardware units [44], our structured convolutions can be easily adapted for memory and compute limited devices.

Compared to the baseline models, the Struct-A versions of ResNets are $2\times$ smaller, while maintaining less than $0.65\%$ loss in accuracy. The more aggressive Struct-B ResNets achieve 60-70% model size reduction with about 2-3% accuracy drop. Compared to other methods, Struct-56-A is $0.75\%$ better than AMC-Res56 [11] of similar complexity and Struct-20-A exceeds ShiftResNet20-6 [42] by $0.45\%$ while being significantly smaller. Similar trends are observed with Struct-Res18 and Struct-Res50 on ImageNet. Struct-56-A and Struct-50-A achieve competitive performance as compared to the recent GhostNets [7]. For MobileNetV2 which is already designed to be efficient, Struct-MV2-A achieves further reduction in multiplications and model size with SOTA performance compared to other methods, see Table 2. Applying structured convolutions to EfficientNet-B1 results in Struct-EffNet that has comparable performance to EfficientNet-B0, as can be seen in Table 4.

The ResNet Struct-A versions have similar number of adds and multiplies (except ResNet50) because, as noted in Sec. 4.2, the sum-pooling contribution is amortized. But sum-pooling starts dominating as the compression gets more aggressive, as can be seen in the number of adds for Struct-B versions. Notably, both "A" and "B" versions of MobileNetV2 observe a dominance of the sum-pooling component. This is because the number of output channels are not enough to amortize the sum-pooling component resulting from the decomposition of the pointwise ($1\times1$ conv) layers.

Fig. 6 compares our method with state-of-the-art structured compression methods - WeightSVD [48], Channel Pruning [12], and Tensor-train [35]. Note, the results were obtained from [19]. Our proposed method achieves approximately $1\%$ improvement over the second best method for ResNet18 ($2\times$) and MobileNetV2 ($1.3\times$). Especially for MobileNetV2, this improvement is valuable since it significantly outperforms all the other methods (see Struct-V2-A in Table 2).

## 6.2 Semantic Segmentation

Table 5: Evaluation of proposed method on Cityscapes [4] using HRNetV2.

| HRNetV2-W18 -Small-v2 | #adds ($\times10^9$) | #mults ($\times10^9$) | #params ($\times10^6$) | Mean IoU (in %) |
|---|---|---|---|---|
| Original | 76.8 | 77.1 | 3.9 | 76.1 |
| Struct-HR-A | 54.3 | 54.0 | 1.9 | 74.6 |

After demonstrating the superiority of our method on image classification, we evaluate it for semantic segmentation that requires reproducing fine details around object boundaries. We apply our method to a recently developed state-of-the-art HRNet [39]. Table 5 shows that the structured convolutions can significantly improve our segmentation model efficiency: HRNet model is reduced by 50% in size, and 30% in number of additions and multiplications, while having only 1.5% drop in mIoU. More results for semantic segmentation can be found in the supplementary material.

### 6.3 Computational overhead of Structural Regularization term

The proposed method involves computing the Structural Regularization term (3) during training. Although the focus of this work is more on inference efficiency, we measure the memory and time-per-iteration for training *with* and *w/o* the Structural Regularization loss on an NVIDIA V100 GPU. We report these numbers for Struct-18-A and Struct-MV2-A architectures below. As observed, the additional computational cost of the regularization term is negligible for a batchsize of 256. This is because, mathematically, the regularization term, $\sum_{l=1}^{L} \frac{\left\| (I - A_l A_l^+) W_l \right\|_F}{\|W_l\|_F}$, is independent of the input size. Hence, when using a large batchsize for training, the reglularization term's memory and runtime overhead is relatively small (less than $5\%$ for Struct-MV2-A and $10\%$ for Struct-18-A).

Table 6: Memory and runtime costs of training *with* and *w/o* Structural Regularization (SR) loss

| Training costs (batchsize=256) | Struct-18-A | | Struct-MV2-A | |
|---|---|---|---|---|
| | Mem | seconds / iter | Mem | seconds / iter |
| With SR loss | 9.9GB | 0.46s | 18.8GB | 0.38s |
| Without SR loss | 9.2GB | 0.44s | 17.9GB | 0.37s |

### 6.4 Directly training Structured Convolutions as an architectural feature

In our proposed method, we train the architecture with original $C \times N \times N$ kernels in place and the regularization loss imposes desired structure on these kernels. At the end of training, we decompose the $C \times N \times N$ kernels and replace each with a sum-pooling layer and smaller layer of $c \times n \times n$ kernels.

A more *direct* approach could be to train the decomposed architecture (with the sum-pooling + $c \times n \times n$ layers in place) directly. The regularization term is not required in this direct approach, as there is no decomposition step, hence eliminating the computation overhead shown in Table 6. We experimented with this direct training method and observed that the regularization based approach always outperformed the direct approach (by $1.5\%$ for Struct-18-A and by $0.9\%$ for Struct-MV2-A). This is because, as pointed out in Sec. 5.1, the direct method optimizes the weights in a restricted subspace of $c \times n \times n$ kernels right from the start, whereas the regularization based approach gradually moves the weights from the larger ($C \times N \times N$) subspace to the restricted subspace with gradual imposition of the structure constraints via the regularization loss.

## 7 Conclusion

In this work, we propose Composite Kernels and Structured Convolutions in an attempt to exploit redundancy in the *implicit* structure of convolution kernels. We show that Structured Convolutions can be decomposed into a computationally cheap sum-pooling component followed by a significantly smaller convolution, by training the model using an intuitive structural regularization loss. The effectiveness of the proposed method is demonstrated via extensive experiments on image classification and semantic segmentation benchmarks. Sum-pooling relies purely on additions, which are known to be extremely power-efficient. Hence, our method shows promise in deploying deep models on low-power devices. Since our method keeps the convolutional structures, it allows integration of further model compression schemes, which we leave as future work.

## Broader Impact

The method proposed in this paper promotes the adaption of deep learning neural networks into memory and compute limited devices, allowing a broader acceptance of machine learning solutions for a spectrum of real-life use cases. By reducing the associated hardware costs of the neural network systems, it aims at making such technology affordable to larger communities. It empowers people by facilitating access to the latest developments in this discipline of science. It neither leverages biases in data nor demands user consent for the use of data.

## Acknowledgements and Funding Disclosure

We would like to thank our Qualcomm AI Research colleagues for their support and assistance, in particular that of Andrey Kuzmin, Tianyu Jiang, Khoi Nguyen, Kwanghoon An and Saurabh Pitre.

Funding in direct support of this work: Qualcomm Technologies, Inc.

## Footnotes

*Qualcomm AI Research is an initiative of Qualcomm Technologies, Inc.

[2]Our Efficientnet reproduction of baselines (B0 and B1) give results slightly inferior to [37]. Our Struct-EffNet is created on top of this EfficientNet-B1 baseline.

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
