[Supplementary Material]

# Supplementary Material: 'Structured Convolutions for Efficient Neural Network Design'

**Yash Bhalgat**      **Yizhe Zhang**      **Jamie Menjay Lin**      **Fatih Porikli**

Qualcomm AI Research[*]

{ybhalgat, yizhez, jmlin, fporikli}@qti.qualcomm.com

## A    Appendix

### A.1    Structured Convolutions with arbitrary Padding, Stride and Dilation

In the main paper, we showed that a Structured Convolution can be decomposed into a Sum-Pooling component followed by a smaller convolution operation with a kernel composed of the $\alpha$'s. In this section, we discuss how to calculate the equivalent stride, padding and dilation needed for the resulting decomposed sum-pooling and convolution operations.

#### A.1.1    Padding

The easiest of these three attributes is *padding*. Fig. 1 shows an example of a structured convolution with a $3 \times 3$ kernel (i.e. $N = 3$) with underlying parameter $n = 2$. Hence, it can be decomposed into a $2 \times 2$ sum-pooling operation followed by a $2 \times 2$ convolution. As shown in the figure, to preserve the same output after the decomposition, the sum-pooling component should use the same padding as the *original* $3 \times 3$ convolution, whereas the smaller $2 \times 2$ convolution is performed without padding.

This leads us to a more general result that - if the original convolution uses a padding of $p$, then, after the decomposition, the sum-pooling should be performed with padding $p$ and the smaller convolution (with $\alpha$'s) should be performed without padding.

#### A.1.2    Stride

The above rule can be simply extended to the case where the original $3 \times 3$ structured convolution has a stride associated with it. The general rule is - if the original convolution uses a stride of $s$, then, after the decomposition, the sum-pooling should be performed with a stride of 1 and the smaller convolution (with $\alpha$'s) should be performed with a stride of $s$.

#### A.1.3    Dilation

Dilated or atrous convolutions are prominent in semantic segmentation architectures. Hence, it is important to consider how we can decompose dilated structured convolutions. Fig. 2 shows an example of a $3 \times 3$ structured convolution with a dilation of 2. As can be seen in the figure, to preserve the same output after decomposition, *both* the sum-pooling component and the smaller convolution (with $\alpha$'s) has to be performed with a dilation factor same as the original convolution.

Fig. 3 summarizes the aforementioned rules regarding padding, stride and dilation.

---

[*]Qualcomm AI Research is an initiative of Qualcomm Technologies, Inc.

Figure 1: Decomposition of a $3 \times 3$ Structured Convolution with a padding of 1. Top shows the conventional operation of the convolution. Bottom shows the equivalent operation using sum-pooling.

Figure 2: Decomposition of a $3 \times 3$ Structured Convolution with a dilation of 2. Top shows the conventional operation of the convolution. Bottom shows the equivalent operation using sum-pooling.

## A.2    Training Implementation Details

**Image Classification.**    For both ImageNet and CIFAR-10 benchmarks, we train all the ResNet architectures from scratch with the Structural Regularization (SR) loss. We set $\lambda$ to 0.1 for the Struct-A versions and 1.0 for the Struct-B versions throughout training. For MobileNetV2, we first train the deep network from scratch *without* SR loss (i.e. $\lambda = 0$) for 90 epochs to obtain pretrained weights and then apply SR loss with $\lambda = 1.0$ for further 150 epochs. For EfficientNet-B0, we first train without SR loss for 90 epochs and then apply SR loss with $\lambda = 1.0$ for further 250 epochs.

For CIFAR-10, we train the ResNets for 200 epochs using a batch size of 128 and an initial learing rate of 0.1 which is decayed by a factor of 10 at 80 and 120 epochs. We use a weight decay of 0.0001 throughout training. On ImageNet, we use a cosine learning rate schedule with an SGD optimizer for

Figure 3: Decomposition of a general structured convolution with stride, padding and dilation. The blocked arrows indicate the dimensions of the input and output tensors. Top shows the conventional operation of the convolution. Bottom shows the equivalent operation using sum-pooling.

training all architectures. We train the ResNets using a batch size of 256 and weight decay of 0.0001 for 200 epochs starting with an initial learning rate of 0.1.

For MobileNetV2, we use a weight decay of 0.00004 and batch size 128 throughout training. In the first phase (with $\lambda = 0$), we use an initial learning rate of 0.5 for 90 epochs and in the second phase, we start a new cosine schedule with an initial learning rate of 0.1 for the next 150 epochs. We train EfficientNet-B0 using Autoaugment, a weight decay of 0.00004 and batch size 384. We use an initial learning rate of 0.5 in the first phase and we start a new cosine schedule for the second phase with an initial learning rate of 0.1 for the next 250 epochs.

**Semantic Segmentation.** For training Struct-HRNet-A on Cityscapes, we start from a pre-trained HRNet model and train using structural regularization loss. We set $\lambda$ to 1.0. We use a cosine learning rate schedule with an initial learning rate of 0.01. The use image resolution of $1024 \times 2048$ for training, same as the original image size. We train for 90000 iterations using a batch size of 4.

We show additional results with PSPNet in Sec. A.3 below. We follow a similar training process for training Struct-PSPNet-A where we start from a pre-trained PSPNet101 [5].

### A.3 Additional results on Semantic Segmentation

In Table 1 and 2, we present additional results for HRNetV2-W18-Small-v1 [3] (note this is different from HRNetV2-W18-Small-v2 reported in the main paper) and PSPNet101 [5] on Cityscapes dataset.

Table 1: Evaluation of proposed method on Cityscapes using HRNetV2-W18-Small-v1 [3].

| HRNetV2-W18 -Small-v1 | #adds ($\times 10^9$) | #mults ($\times 10^9$) | #params ($\times 10^6$) | mIoU (in %) |
|---|---|---|---|---|
| Original | 33.4 | 33.6 | 1.5 | 70.3 |
| Struct-HR-A-V1 | 24.6 | 24.5 | 0.77 | 67.5 |

Table 2: Evaluation of proposed method on Cityscapes using PSPNet101 [5].

| PSPNet101 | #adds ($\times 10^9$) | #mults ($\times 10^9$) | #params ($\times 10^6$) | mIoU (in %) |
|---|---|---|---|---|
| Original | 2094 | 2096 | 68.1 | 79.3 |
| Struct-PSP-A | 1325 | 1327 | 43.0 | 76.6 |

## A.4 Layer-wise compression ratios for compared architectures

As mentioned in the Experiments section of the main paper, we use non-uniform selection for the per-layer compression ratios ($CN^2/cn^2$) for MobileNetV2 and EfficientNet-B0 as well as HRNet for semantic segmentation. Tables 6 and 7 show the layerwise $\{c, n\}$ parameters for each layer of the Struct-MV2-A and Struct-MV2-B architectures. Table 8 shows these per-layer $\{c, n\}$ parameters for Struct-EffNet.

For Struct-HRNet-A, we apply Structured Convolutions only in the spatial dimension, i.e. we use $c = C$, hence there's no decomposition across the channel dimension. For 3×3 convolutional kernels, we use $n = 2$, which means a 3×3 convolution is decomposed into a 2×2 sum pooling followed by a 2×2 convolution. And for 1×1 convolutions, where $N = 1$, we use $n = 1$ which is the only possiblility for $n$ since $1 \leq n \leq N$. We do not use Structured Convolutions in the initial two convolution layers and last convolution layer.

For Struct-PSPNet, similar to Struct-HRNet-A, we apply use structured convolutions in all the convolution layers except the first and last layer. For 3×3 convolutions, the structured convolution uses $c = C$ and $n = 2$. For 1×1 convolutions, the structured convolution uses $c = round(2 \times C/3)$ and $n = 1$.

## A.5 Sensitivity of Structural Regularization w.r.t $\lambda$

In Sec. 5.1, we introduced the Structural Regularization (SR) loss and proposed to train the network using this regularization with a weight $\lambda$. In this section, we investigate the variation in the final performance of the model (after decomposition) when trained with different values of $\lambda$.

We trained Struct-Res18-A and Struct-Res18-B with different values of $\lambda$. Note that when training both "A" and "B" versions, we start with the original architecture for ResNet18 and train it from scratch with the SR loss. After this first step, we then decompose the weights using $\alpha_l = A^+ W_l$ to get the decomposed architecture. Tables 3 and 4 show the accuracy of Struct-Res18-A and Struct-Res18-B both pre-decomposition and post-decomposition.

Table 3: ImageNet performance of Struct-Res18-A trained with different $\lambda$

| $\lambda$ | Acc. (before decomposition) | Top-1 Acc. (after decomposition) |
|---|---|---|
| 1.0 | 69.08% | 69.11% |
| 0.5 | 69.21% | 69.09% |
| 0.1 | 69.17% | **69.13**% |
| 0.05 | 69.16% | 69.05% |
| 0.01 | 69.20% | 69.09% |
| 0.001 | 69.23% | 68.57% |

Table 4: ImageNet performance of Struct-Res18-B trained with different $\lambda$

| $\lambda$ | Acc. (before decomposition) | Top-1 Acc. (after decomposition) |
|---|---|---|
| 1.0 | 66.04% | **66.19**% |
| 0.5 | 66.11% | 66.01% |
| 0.1 | 66.01% | 65.89% |
| 0.05 | 65.97% | 65.65% |
| 0.01 | 65.99% | 64.47% |
| 0.001 | 65.91% | 58.19% |

Table 5: ImageNet performance *before* and *after* decomposition for other architectures

| Architecture | Acc. (before decomposition) | Top-1 Acc. (after decomposition) |
|---|---|---|
| Struct-50-A | 75.68% | 75.65% |
| Struct-50-B | 73.52% | 73.41% |
| Struct-V2-A | 71.33% | 71.29% |
| Struct-V2-B | 65.01% | 64.93% |
| Struct-EffNet | 76.59% | 76.40% |

From Table 3, we can see that the accuracy after decomposition isn't affected much by the choice of $\lambda$. When $\lambda$ varies from 0.01 to 1.0, the post-decomposition accuracy only changed by 0.08%. Similar trends are observed in Table 4 when we are compressing more aggressively. But the sensitivity of the performance w.r.t. $\lambda$ is slightly higher in the "B" version. Also, we can see that when $\lambda = 0.001$, the difference between pre-decomposition and post-decomposition accuracy is significant. Since $\lambda$ is

very small in this case, the Structural Regularization loss does not impose the desired structure on the convolution kernels effectively. As a result, after decomposition, it leads to a loss in accuracy.

In Table 5, we show the ImageNet performance of other architectures (from Tables 1, 2, 3, 4 of main paper) before and after the decomposition is applied.

## A.6 Expressive power of the Sum-Pooling component

To show that the sum-pooling layers indeed capture meaningful features, we perform a toy experiment where we swap all $3 \times 3$ depthwise convolution kernels in MobileNetV2 with $2 \times 2$ kernels and train the architecture. We observed that this leads to a severe performance degradation of $4.5\%$ compared to the Struct-V2-A counterpart. This, we believe, is due to the loss of receptive field that was being captured by the sum-pooling part of structured convolutions.

## A.7 Inference Latency

In Sec. 6.1 of the paper, we pointed out that the actual inference time of our method depends on how the software optimizations and scheduling are implemented. Additions are much faster and power efficient than multiplications on low-level hardware [1, 2]. However, this is not exploited on conventional platforms like GPUs which use FMA (Fused Multiply-Add) instructions. Considering hardware accelerator implementations [4] for sum-pooling, the theoretical gains of structured convolutions can be realized. We provide *estimates* for the latencies based on measurements on a Intel Xeon CPU W-2123 platform assuming that the software optimizations and scheduling for the sum-pooling operation are implemented. Please refer the table below.

| ResNet18 | 0.039s | MobilenetV2 | 0.088s | EfficientNet-B1 | 0.114s |
|---|---|---|---|---|---|
| Struct-18-A | **0.030s** | Struct-MV2-A | **0.078s** | Struct-EffNet | **0.101s** |

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

Table 6: Layerwise $\{c, n\}$ configuration for Struct-MV2-A architecture

| Idx | Dimension $C_{out} \times C \times N \times N$ | $c$ | $n$ |
|---|---|---|---|
| 1 | $32 \times 3 \times 3 \times 3$ | 3 | 3 |
| 2 | $32 \times 1 \times 3 \times 3$ | 1 | 3 |
| 3 | $16 \times 32 \times 1 \times 1$ | 32 | 1 |
| 4 | $96 \times 16 \times 1 \times 1$ | 16 | 1 |
| 5 | $96 \times 1 \times 3 \times 3$ | 1 | 3 |
| 6 | $24 \times 96 \times 1 \times 1$ | 96 | 1 |
| 7 | $144 \times 24 \times 1 \times 1$ | 24 | 1 |
| 8 | $144 \times 1 \times 3 \times 3$ | 1 | 3 |
| 9 | $24 \times 144 \times 1 \times 1$ | 144 | 1 |
| 10 | $144 \times 24 \times 1 \times 1$ | 24 | 1 |
| 11 | $144 \times 1 \times 3 \times 3$ | 1 | 3 |
| 12 | $32 \times 144 \times 1 \times 1$ | 144 | 1 |
| 13 | $192 \times 32 \times 1 \times 1$ | 32 | 1 |
| 14 | $192 \times 1 \times 3 \times 3$ | 1 | 3 |
| 15 | $32 \times 192 \times 1 \times 1$ | 192 | 1 |
| 16 | $192 \times 32 \times 1 \times 1$ | 32 | 1 |
| 17 | $192 \times 1 \times 3 \times 3$ | 1 | 3 |
| 18 | $32 \times 192 \times 1 \times 1$ | 192 | 1 |
| 19 | $192 \times 32 \times 1 \times 1$ | 32 | 1 |
| 20 | $192 \times 1 \times 3 \times 3$ | 1 | 3 |
| 21 | $64 \times 192 \times 1 \times 1$ | 192 | 1 |
| 22 | $384 \times 64 \times 1 \times 1$ | 64 | 1 |
| 23 | $384 \times 1 \times 3 \times 3$ | 1 | 3 |
| 24 | $64 \times 384 \times 1 \times 1$ | 384 | 1 |
| 25 | $384 \times 64 \times 1 \times 1$ | 64 | 1 |
| 26 | $384 \times 1 \times 3 \times 3$ | 1 | 3 |
| 27 | $64 \times 384 \times 1 \times 1$ | 384 | 1 |
| 28 | $384 \times 64 \times 1 \times 1$ | 64 | 1 |
| 29 | $384 \times 1 \times 3 \times 3$ | 1 | 3 |
| 30 | $64 \times 384 \times 1 \times 1$ | 384 | 1 |
| 31 | $384 \times 64 \times 1 \times 1$ | 64 | 1 |
| 32 | $384 \times 1 \times 3 \times 3$ | 1 | 3 |
| 33 | $96 \times 384 \times 1 \times 1$ | 384 | 1 |
| 34 | $576 \times 96 \times 1 \times 1$ | 96 | 1 |
| 35 | $576 \times 1 \times 3 \times 3$ | 1 | 3 |
| 36 | $96 \times 576 \times 1 \times 1$ | 576 | 1 |
| 37 | $576 \times 96 \times 1 \times 1$ | 96 | 1 |
| 38 | $576 \times 1 \times 3 \times 3$ | 1 | 3 |
| 39 | $96 \times 576 \times 1 \times 1$ | 576 | 1 |
| 40 | $576 \times 96 \times 1 \times 1$ | 96 | 1 |
| 41 | $576 \times 1 \times 3 \times 3$ | 1 | 3 |
| 42 | $160 \times 576 \times 1 \times 1$ | 576 | 1 |
| 43 | $960 \times 160 \times 1 \times 1$ | 160 | 1 |
| 44 | $960 \times 1 \times 3 \times 3$ | 1 | 3 |
| 45 | $160 \times 960 \times 1 \times 1$ | 960 | 1 |
| 46 | $960 \times 160 \times 1 \times 1$ | 160 | 1 |
| 47 | $960 \times 1 \times 3 \times 3$ | 1 | 3 |
| 48 | $160 \times 960 \times 1 \times 1$ | 960 | 1 |
| 49 | $960 \times 160 \times 1 \times 1$ | 160 | 1 |
| 50 | $960 \times 1 \times 3 \times 3$ | 1 | 3 |
| 51 | $320 \times 960 \times 1 \times 1$ | 840 | 1 |
| 52 | $1280 \times 320 \times 1 \times 1$ | 160 | 1 |
| classifier | $1000 \times 1280 \times 1 \times 1$ | 640 | 1 |

Table 7: Layerwise $\{c, n\}$ configuration for Struct-MV2-B architecture

| Idx | Dimension $C_{out} \times C \times N \times N$ | $c$ | $n$ |
|---|---|---|---|
| 1 | $32 \times 3 \times 3 \times 3$ | 3 | 3 |
| 2 | $32 \times 1 \times 3 \times 3$ | 1 | 3 |
| 3 | $16 \times 32 \times 1 \times 1$ | 32 | 1 |
| 4 | $96 \times 16 \times 1 \times 1$ | 16 | 1 |
| 5 | $96 \times 1 \times 3 \times 3$ | 1 | 3 |
| 6 | $24 \times 96 \times 1 \times 1$ | 48 | 1 |
| 7 | $144 \times 24 \times 1 \times 1$ | 12 | 1 |
| 8 | $144 \times 1 \times 3 \times 3$ | 1 | 3 |
| 9 | $24 \times 144 \times 1 \times 1$ | 72 | 1 |
| 10 | $144 \times 24 \times 1 \times 1$ | 12 | 1 |
| 11 | $144 \times 1 \times 3 \times 3$ | 1 | 3 |
| 12 | $32 \times 144 \times 1 \times 1$ | 72 | 1 |
| 13 | $192 \times 32 \times 1 \times 1$ | 16 | 1 |
| 14 | $192 \times 1 \times 3 \times 3$ | 1 | 3 |
| 15 | $32 \times 192 \times 1 \times 1$ | 96 | 1 |
| 16 | $192 \times 32 \times 1 \times 1$ | 16 | 1 |
| 17 | $192 \times 1 \times 3 \times 3$ | 1 | 2 |
| 18 | $32 \times 192 \times 1 \times 1$ | 96 | 1 |
| 19 | $192 \times 32 \times 1 \times 1$ | 16 | 1 |
| 20 | $192 \times 1 \times 3 \times 3$ | 1 | 2 |
| 21 | $64 \times 192 \times 1 \times 1$ | 96 | 1 |
| 22 | $384 \times 64 \times 1 \times 1$ | 32 | 1 |
| 23 | $384 \times 1 \times 3 \times 3$ | 1 | 2 |
| 24 | $64 \times 384 \times 1 \times 1$ | 192 | 1 |
| 25 | $384 \times 64 \times 1 \times 1$ | 32 | 1 |
| 26 | $384 \times 1 \times 3 \times 3$ | 1 | 2 |
| 27 | $64 \times 384 \times 1 \times 1$ | 192 | 1 |
| 28 | $384 \times 64 \times 1 \times 1$ | 32 | 1 |
| 29 | $384 \times 1 \times 3 \times 3$ | 1 | 2 |
| 30 | $64 \times 384 \times 1 \times 1$ | 192 | 1 |
| 31 | $384 \times 64 \times 1 \times 1$ | 32 | 1 |
| 32 | $384 \times 1 \times 3 \times 3$ | 1 | 2 |
| 33 | $96 \times 384 \times 1 \times 1$ | 192 | 1 |
| 34 | $576 \times 96 \times 1 \times 1$ | 48 | 1 |
| 35 | $576 \times 1 \times 3 \times 3$ | 1 | 2 |
| 36 | $96 \times 576 \times 1 \times 1$ | 288 | 1 |
| 37 | $576 \times 96 \times 1 \times 1$ | 48 | 1 |
| 38 | $576 \times 1 \times 3 \times 3$ | 1 | 2 |
| 39 | $96 \times 576 \times 1 \times 1$ | 288 | 1 |
| 40 | $576 \times 96 \times 1 \times 1$ | 48 | 1 |
| 41 | $576 \times 1 \times 3 \times 3$ | 1 | 2 |
| 42 | $160 \times 576 \times 1 \times 1$ | 288 | 1 |
| 43 | $960 \times 160 \times 1 \times 1$ | 80 | 1 |
| 44 | $960 \times 1 \times 3 \times 3$ | 1 | 2 |
| 45 | $160 \times 960 \times 1 \times 1$ | 480 | 1 |
| 46 | $960 \times 160 \times 1 \times 1$ | 80 | 1 |
| 47 | $960 \times 1 \times 3 \times 3$ | 1 | 2 |
| 48 | $160 \times 960 \times 1 \times 1$ | 480 | 1 |
| 49 | $960 \times 160 \times 1 \times 1$ | 80 | 1 |
| 50 | $960 \times 1 \times 3 \times 3$ | 1 | 3 |
| 51 | $320 \times 960 \times 1 \times 1$ | 480 | 1 |
| 52 | $1280 \times 320 \times 1 \times 1$ | 160 | 1 |
| classifier | $1000 \times 1280 \times 1 \times 1$ | 560 | 1 |

Table 8: Layerwise $\{c, n\}$ configuration for Struct-EffNet architecture

| Idx | Dimension $C_{out} \times C \times N \times N$ | $c$ | $n$ |
|---|---|---|---|
| 1 | $32 \times 3 \times 3 \times 3$ | 3 | 3 |
| 2 | $32 \times 1 \times 3 \times 3$ | 1 | 3 |
| 3 | $16 \times 32 \times 1 \times 1$ | 32 | 1 |
| 4 | $16 \times 1 \times 3 \times 3$ | 1 | 3 |
| 5 | $16 \times 16 \times 1 \times 1$ | 16 | 1 |
| 6 | $96 \times 16 \times 1 \times 1$ | 16 | 1 |
| 7 | $96 \times 1 \times 3 \times 3$ | 1 | 3 |
| 8 | $24 \times 96 \times 1 \times 1$ | 96 | 1 |
| 9 | $144 \times 24 \times 1 \times 1$ | 24 | 1 |
| 10 | $144 \times 1 \times 3 \times 3$ | 1 | 3 |
| 11 | $24 \times 144 \times 1 \times 1$ | 144 | 1 |
| 12 | $144 \times 24 \times 1 \times 1$ | 24 | 1 |
| 13 | $144 \times 1 \times 3 \times 3$ | 1 | 3 |
| 14 | $24 \times 144 \times 1 \times 1$ | 144 | 1 |
| 15 | $144 \times 24 \times 1 \times 1$ | 24 | 1 |
| 16 | $144 \times 1 \times 5 \times 5$ | 1 | 5 |
| 17 | $40 \times 144 \times 1 \times 1$ | 144 | 1 |
| 18 | $240 \times 40 \times 1 \times 1$ | 40 | 1 |
| 19 | $240 \times 1 \times 5 \times 5$ | 1 | 5 |
| 20 | $40 \times 240 \times 1 \times 1$ | 240 | 1 |
| 21 | $240 \times 40 \times 1 \times 1$ | 40 | 1 |
| 22 | $240 \times 1 \times 5 \times 5$ | 1 | 5 |
| 23 | $40 \times 240 \times 1 \times 1$ | 240 | 1 |
| 24 | $240 \times 40 \times 1 \times 1$ | 40 | 1 |
| 25 | $240 \times 1 \times 3 \times 3$ | 1 | 3 |
| 26 | $80 \times 240 \times 1 \times 1$ | 240 | 1 |
| 27 | $480 \times 80 \times 1 \times 1$ | 64 | 1 |
| 28 | $480 \times 1 \times 3 \times 3$ | 1 | 3 |
| 29 | $80 \times 480 \times 1 \times 1$ | 360 | 1 |
| 30 | $480 \times 80 \times 1 \times 1$ | 64 | 1 |
| 31 | $480 \times 1 \times 3 \times 3$ | 1 | 3 |
| 32 | $80 \times 480 \times 1 \times 1$ | 360 | 1 |
| 33 | $480 \times 80 \times 1 \times 1$ | 64 | 1 |
| 34 | $480 \times 1 \times 3 \times 3$ | 1 | 3 |
| 35 | $80 \times 480 \times 1 \times 1$ | 360 | 1 |
| 36 | $480 \times 80 \times 1 \times 1$ | 64 | 1 |
| 37 | $480 \times 1 \times 5 \times 5$ | 1 | 5 |
| 38 | $112 \times 480 \times 1 \times 1$ | 360 | 1 |
| 39 | $672 \times 112 \times 1 \times 1$ | 80 | 1 |
| 40 | $672 \times 1 \times 5 \times 5$ | 1 | 5 |
| 41 | $112 \times 672 \times 1 \times 1$ | 560 | 1 |
| 42 | $672 \times 112 \times 1 \times 1$ | 96 | 1 |
| 43 | $672 \times 1 \times 5 \times 5$ | 1 | 5 |
| 44 | $112 \times 672 \times 1 \times 1$ | 560 | 1 |
| 45 | $672 \times 112 \times 1 \times 1$ | 96 | 1 |
| 46 | $672 \times 1 \times 5 \times 5$ | 1 | 5 |
| 47 | $112 \times 672 \times 1 \times 1$ | 560 | 1 |
| 48 | $672 \times 112 \times 1 \times 1$ | 96 | 1 |
| 49 | $672 \times 1 \times 5 \times 5$ | 1 | 5 |
| 50 | $192 \times 672 \times 1 \times 1$ | 560 | 1 |
| 51 | $1152 \times 192 \times 1 \times 1$ | 100 | 1 |
| 52 | $1152 \times 1 \times 5 \times 5$ | 1 | 5 |
| 53 | $192 \times 1152 \times 1 \times 1$ | 640 | 1 |
| 54 | $1152 \times 192 \times 1 \times 1$ | 100 | 1 |
| 55 | $1152 \times 1 \times 5 \times 5$ | 1 | 5 |
| 56 | $192 \times 1152 \times 1 \times 1$ | 640 | 1 |
| 57 | $1152 \times 192 \times 1 \times 1$ | 100 | 1 |
| 58 | $1152 \times 1 \times 5 \times 5$ | 1 | 5 |
| 59 | $192 \times 1152 \times 1 \times 1$ | 640 | 1 |
| 60 | $1152 \times 192 \times 1 \times 1$ | 100 | 1 |
| 61 | $1152 \times 1 \times 5 \times 5$ | 1 | 5 |
| 62 | $192 \times 1152 \times 1 \times 1$ | 576 | 1 |
| 63 | $1152 \times 192 \times 1 \times 1$ | 160 | 1 |
| 64 | $1152 \times 1 \times 3 \times 3$ | 1 | 3 |
| 65 | $320 \times 1152 \times 1 \times 1$ | 576 | 1 |
| 66 | $1920 \times 320 \times 1 \times 1$ | 160 | 1 |
| 67 | $1920 \times 1 \times 3 \times 3$ | 1 | 3 |
| 68 | $320 \times 1920 \times 1 \times 1$ | 960 | 1 |
| 69 | $1280 \times 320 \times 1 \times 1$ | 160 | 1 |
| classifier | $1000 \times 1280 \times 1 \times 1$ | 480 | 1 |