[Reviews · NeurIPS 2020]

Review 1

Summary and Contributions: The paper presents a new type of structured convolution using sum pooling followed by a smaller convolution to maintain the receptive field (i.e. effective kernel size) while using fewer operations (addition and multiplication) and parameters.

Strengths: The concept of the structured convolution proposed in the paper is overall novel and relevant to the NeurIPS community for its potential applications.

Weaknesses: The major weaknesses of this paper are the soundness and significance. In terms of soundness, the biggest issue is the way the empirical evaluation was designed and executed. Firstly, it's extremely odd that the authors decided to implement the structured convolution as a post-training conversion, rather than directly implementing the structured convolution as a trainable architectural feature (e.g. like depthwise convolution), which can eliminate the computational cost of the regularization term in Eq (3) (likely expensive but unacceptably undiscussed in the paper) and may further improve the accuracy results. Otherwise, the authors should demonstrate the feasibility of performing Eq (3) as quick fine-tuning for fully trained (regular) networks to justify their approach. Secondly, although the reduction in numbers of operations and parameters is overall impressive, the decrease in accuracies compared to baselines (Table 1 to 5) is also non-negligible, especially for more recent networks like EfficientNet and MobileNetV2. Even though the proposed approach compares favorably to other compression methods [48, 12, 35] in Fig 6, the significance of the margins isn't clear without standard errors shown in the figure. More importantly, [48, 12, 35] are all relatively dated, making the comparison questionable. Finally, the authors should include actual inference speed on generally available devices (i.e. CPU or GPU) using reasonably recent DL frameworks for readers' reference. In terms of significance, given the less than satisfactory empirical evaluation as stated above, the paper is unfortunately not significant enough.

Correctness: The paper is overall correct.

Clarity: The paper is well written.

Relation to Prior Work: The related work section is overall fine, although it's unclear if the authors are comparing against SOTA in Fig 6, arguably the most important part among all results.

Reproducibility: Yes

Additional Feedback: [Reply to Rebuttal] I appreciate the authors' detailed feedback and have upgraded my score to 7. Below are some more suggestions. 1. The accuracy drop due to using the structured convolution as an architectural feature in training (i.e. the direct approach) is unexpectedly high. Please discuss more or consider fixing the issue (for the obvious value of accelerating training too). 2. Please do include the analysis of the regularizer and discuss its pros and cons (e.g. when batch size is small or convolution kernel is large). 3. Consider working with hardware experts to provide readers with estimated speedups for inference using TPUs or other custom hardware.


Review 2

Summary and Contributions: This paper proposes a network compression method by imposing compositionally of convolution kernels. The proposed method greatly reduces the number of multiplication and the number of parameters in the original network. Comparing to baselines, the proposed method is able to achieve a similar compression ratio while maintaining a slightly higher accuracy.

Strengths: - The idea of structured convolution is quite interesting and reasonable to limit the degrees of freedom for convolution kernels. The idea is also quite simple and can thus generalize to most of CNN architectures. - This paper is well-written. They explained the structured convolutions using both intuitive diagrams and formal maths. - Many experiments are conducted across datasets, tasks and architectures (CIFAR, ImageNet, Cityscapes, ResNet, MobileNet, EfficientNet), showing that the proposed method can empirically generalize well.

Weaknesses: The central idea of structured convolution resembles to using smaller kernel size. It would be better if the authors can explain the difference between simply swapping out 3x3 convs with 2x2 convs (c.f. Fig 1 (b)). I can see that conceptually they are not exactly equivalent, but maybe some experiments should be done to justify that this conceptual difference does make a difference in practice.

Correctness: The claims and method make sense. And the experiments are done properly. A minor issue, at Line 118, should it be 1 <= n <= N and 1<= c <= C?

Clarity: This paper is in general well-written. However, it is a bit confusing in Sec 4 when introducing structured convolutions. Using the number of betas and the numbers of 1s to define a structure is not very intuitive. Although the intuition is mentioned in line 128. It might be better to introduce that first prior to the definition.

Relation to Prior Work: This paper has clearly discussed the related work in an organized way. And it appears that this work is different enough from related work. However, I am not very familiar with related work of this paper.

Reproducibility: Yes

Additional Feedback: Post rebuttal: the rebuttal has addressed my concern and I would like to keep my score of 7.


Review 3

Summary and Contributions: This paper proposed structured convolutions which can be decomposed into a sum-pooling operation and a smaller convolution to achieve higher computation and parameter efficiency.

Strengths: 1. This paper is on the whole clearly written and experimental evaluations are well executed.

Weaknesses: 1. With the imposed structure, the types of features can be extracted might be limited. Would it be possible to approximate some hand-crafted filters, such as 3x3 lapalacian filter, using the proposed structured kernel? 2. There are two steps in the proposed training scheme. Where does the approximation error mainly come from, STEP 1 or Step 2? Could the authors report the model performance after the first step? Why not directly optimize the weights in the restricted subspace? 3. Typo: Figure 3, alpha_2*(x2+x3+x5+x6)

Correctness: Yes

Clarity: Yes.

Relation to Prior Work: I'm not an expert in this field and do not know if the authors have clearly discussed the relationship between the proposed and related works.

Reproducibility: Yes

Additional Feedback: No.


Review 4

Summary and Contributions: This paper introduces a way to decompose convolutional kernels into a sum pooling followed by convolution. This has the advantage of reducing the parameters, additions, and multiplications needed in a convolutional layer, which reduces the computational cost of the network. This does result in slightly lower accuracy, but has the benefit of being more efficient.

Strengths: The paper is very well written and easy to understand. The conceptual decomposition part is clever and explained well. The results are strong, showing the reduction of operations and comparable performance.

Weaknesses: There are a few details that I would like to see discussed more: - What is the training and inference runtime of the network? The paper does correctly mention this relies on the underlying hardware+software, but I'm curious if using current libraries and standard hardware (e.g., GPU, CPU) what the runtime of this approach is. Since efficiency is a main claim, it would be great to see this actually demonstrated (faster runtime, lower power usage, etc), rather than just a reduction of multiplications. - It is a bit unclear how this is actually implemented. At first I thought it was redefining the convolutional operation in the network, but later it the paper it seems like it is using standard convolution then after training it does the decomposition. Unfortunately no code is provided, so it is difficult to understand how this is actually implemented. - Related, how is the 2-step training process done? Is step 1 the standard, full gradient descent approach then step 2 is done once training is finished or are steps 1 and 2 alternated with each iteration of training? Line 207, "The proposed scheme trains the architecture with the original C×N×N kernels in place but with a structural regularization loss." makes it seem like there would be no efficiency gains during training, but only during inference. It also makes it unclear how the efficient version is implemented. If it is using original kernels, adding some details about implementation is critical to actually obtain a reduction of operations.

Correctness: Yes

Clarity: Yes the paper is well written. There are lacking implementation details, which I think are necessary to see that this proposed approach actually reduces operations.

Relation to Prior Work: Yes

Reproducibility: No

Additional Feedback: Overall, I think the paper is good, is well written and makes a good contribution. I think it really needs the implementation details, which will greatly strengthen the paper. --- After rebuttal The rebuttal mostly addressed my concerns. I would have like to see runtime on different hardware (e.g., GPU) to better demonstrate the runtime gains. The results on CPU show a small improvement. It would be great to add results that actually demonstrate this on the targeted hardware, as the authors mentioned "our method is designed for recent accelerators that allow efficient sum-pooling operations, thus the theoretical speedups (Tables 1-4) are realizable on such platforms."

[Author Response · NeurIPS 2020]

1  We thank all reviewers for their valuable feedback. We tried our best to address their questions within the page limit.

2  **(R1,R3)(1):** We agree that R1's suggested approach (of training structured convolution as a architectural feature) is more direct and does not require computing the regularization. We actually experimented with the suggested 'direct' approach and observed that the regularization based approach always outperformed the direct approach (by $1.5\%$ for Struct-18-A and by $0.9\%$ for Struct-MV2-A). We think this is because the direct approach optimizes the weights in a restricted subspace of $c \times n \times n$ kernels right from the start, whereas the regularization based approach gradually moves the weights from the larger ($C \times N \times N$) subspace to the restricted subspace with gradual imposition of the structure constraints using the regularization loss. We will include this analysis and the results in more detail in the revision. We hope that this also addresses **R3's question** "**why not directly optimize the weights in the restricted subspace**?"

10  **(R1)(2):** We agree with R1 that finetuning using Eq (3) from pretrained weights (whenever they are available) may be a useful alternative, but our proposed method also works well with random initialization.

12  **(R1,R4) Computational cost of regularization term:** Although our focus is more on inference efficiency, we measure memory and time-per-iteration for training *with* and *w/o* SR loss, on an NVIDIA V100 GPU. Mathematically, the SR term, $\sum_{l=1}^{L} \frac{\left\| (I - A_l A_l^+) W_l \right\|_F}{\|W_l\|_F}$, is indepen-

| Training costs (batchsize=256) | Struct-18-A | | Struct-MV2-A | |
|---|---|---|---|---|
| | Mem | sec/iter | Mem | sec/iter |
| With SR loss | 9.9GB | 0.46s | 18.8GB | 0.38s |
| Without SR loss | 9.2GB | 0.44s | 17.9GB | 0.37s |

dent of the input size. Hence, when using a large batchsize, the SR term's memory and runtime overhead is relatively small as shown in the table. As R1 suggested, we could eliminate the computational cost of the regularization term if we use the 'direct' approach, but that leads to a noticeable loss in accuracy as shown in above answer.

20  **(R1) Comments on Fig.6**: In Fig. 6, we compare our performance with the widely used *structured compression* approaches [48]-2016, [12]-2017, [35]-2018, since our method belongs to this category. In addition, Tables 1,2,3 present comparisons with some of the most recent methods in efficient architectures (e.g. GhostNet - CVPR 2020, SlimmableNets - ICLR 2019). Following R1's suggestion, we will add error-bars to Fig.6. We show one of the acc v/s parameters plots here for reference. Note that, we provide the **mean $\pm$ stddev** values for our method, whereas the compared methods have only provided their '**best**' numbers. As visible, the proposed method outperforms by a considerable margin ($0.5\%$ than the highly regarded AMC [11]). Looking beyond performance numbers, we think that our proposal of the structured convolutions offers a new way in efficient model design.

31  **(R1,R4) Inference latency:** Following the convention in EfficientNet paper [37], we report the average inference latency over 50 runs measured with batch size 1 on a single core of Intel Xeon CPU W-2123. As discussed on lines 231-235 and also pointed out by R4, our method is designed for recent accelerators [44]

| ResNet18 | 0.039s | MobilenetV2 | 0.088s | EfficientNet-B1 | 0.114s |
|---|---|---|---|---|---|
| Struct-18-A | **0.030s** | Struct-MV2-A | **0.078s** | Struct-EffNet | **0.101s** |

that allow efficient sum-pooling operations, thus the theoretical speedups (Tables 1-4) are realizable on such platforms.

37  **(R3,R4) Details on training Steps 1 and 2:** To answer R4's question, we first train the model *until convergence* in Step1 which imposes the desired structure and then decompose in Step 2. To answer R3's question, the approximation error comes in Step 2 after decomposition ($\alpha_l = A_l^+ W_l$). This is because the desired structure is not enforced, but *induced* via the loss function. We do provide performance numbers after $1^{st}$ step (and $2^{nd}$ step) for ResNet18 in Sec. E (tables 3,4) of supplementary. We will report these numbers for all other architectures too in our camera-ready version.

42  **(R4) Clarification on Implementation Details:** We will add pseudocode and more intuitive diagrams about the training details in the appendix of our camera-ready version. As outlined in Fig. 4 of paper, the implementation steps may be summarized as follows - in Step 1, we train the architecture with original $C \times N \times N$ kernels in place and the regularization loss imposes desired structure on these kernels. Then, in step 2, we decompose the $C \times N \times N$ convolution layer and replace it with a sum-pooling layer followed by a smaller conv layer with kernels of size $c \times n \times n$. The reported parameters and operations (in Tables 1-4) were calculated using https://github.com/sovrasov/flops-counter.pytorch.

48  **(R3) Laplacian filter:** A Laplacian filter can be constructed with the basis ($\beta_i$'s) as shown, hence can be decomposed into just a horizontal

$$\begin{bmatrix} 0 & -1 & 0 \\ -1 & 4 & -1 \\ 0 & -1 & 0 \end{bmatrix} = -1 \times \begin{bmatrix} 0 & 1 & 0 \\ 0 & 1 & 0 \\ 0 & 1 & 0 \end{bmatrix} + -1 \times \begin{bmatrix} 0 & 0 & 0 \\ 1 & 1 & 1 \\ 0 & 0 & 0 \end{bmatrix} + 6 \times \begin{bmatrix} 0 & 0 & 0 \\ 0 & 1 & 0 \\ 0 & 0 & 0 \end{bmatrix}$$

and vertical sum-pooling component. This is an interesting analysis question, that we plan to look into as future work.

51  **(R2):** Simply swapping $3 \times 3$ kernels with $2 \times 2$'s in MobileNetV2 led to a severe drop in accuracy ($\approx 4.5\%$). This, we believe, is due to the loss of receptive field that was being captured by the sum-pooling part of structured convolutions.

53  **(R2,R3) Typos:** Thanks R2, R3 for finding the typos. We will correct them in the camera-ready version.

[Meta-Review · NeurIPS 2020]

Deep net compression by replacing large-kernel convolutions with sum-pooling and smaller convolutions. Following discussion the reviewers are generally positive, and the rebuttal has addressed some of their initial concerns. Overall I think this will make a good poster. The main novelty is the introduction of a useful form of structured convolutions and a demonstration of its applicability for creating efficient neural networks. The results are strong, showing reduction of complexity with comparable performance in terms of accuracy.